

# The complete mitochondrial genome of *Gyps coprotheres* (Aves, Accipitridae, Accipitriformes): phylogenetic analysis of mitogenome among raptors

Emmanuel Oluwasegun Adawaren[1], Morne Du Plessis[2], Essa Suleman[3,6], Duodane Kindler[3], Almero O. Oosthuizen[2], Lillian Mukandiwa[4] and Vinny Naidoo[5]

[1] Department of Paraclinical Science/Faculty of Veterinary Science, University of Pretoria, Pretoria, Gauteng, South Africa
[2] Bioinformatics and Comparative Genomics, South African National Biodiversity Institute, Pretoria, Gauteng, South Africa
[3] Molecular Diagnostics, Council for Scientific and Industrial Research, Pretoria, Gauteng, South Africa
[4] Department of Paraclinical Science/Faculty of Veterinary Science, University of Pretoria, South Africa
[5] Paraclinical Science/Faculty of Veterinary Science, University of Pretoria, Pretoria, Gauteng, South Africa
[6] Current affiliation: Bioinformatics and Comparative Genomics, South African National Biodiversity Institute, Pretoria, Gauteng, South Africa

## ABSTRACT

Three species of Old World vultures on the Asian peninsula are slowly recovering from the lethal consequences of diclofenac. At present the reason for species sensitivity to diclofenac is unknown. Furthermore, it has since been demonstrated that other Old World vultures like the Cape (*Gyps coprotheres*; CGV) and griffon (*G. fulvus*) vultures are also susceptible to diclofenac toxicity. Oddly, the New World Turkey vulture (*Cathartes aura*) and pied crow (*Corvus albus*) are not susceptible to diclofenac toxicity. As a result of the latter, we postulate an evolutionary link to toxicity. As a first step in understanding the susceptibility to diclofenac toxicity, we use the CGV as a model species for phylogenetic evaluations, by comparing the relatedness of various raptor species known to be susceptible, non-susceptible and suspected by their relationship to the Cape vulture mitogenome. This was achieved by next generation sequencing and assembly. The Cape vulture mitogenome had a genome size of 16,908 bp. The mitogenome phylogenetic analysis indicated a close evolutionary relationship between Old World vultures and other members of the Accipitridae as indicated by bootstrap value of 100% on the phylogenetic trees. Based on this, we postulate that the other species could also be sensitive to the toxic effects of diclofenac. This warrants further investigations.

## INTRODUCTION

Generally, vultures may be classified as Old or New World vultures based on the apparent convergent evolutionary scavenging feeding habit (*Seibold & Helbig, 1995*). However, in

Corresponding author
Emmanuel Oluwasegun Adawaren, adawarenvet1@yahoo.com

reality, they are a morphologically and evolutionary diverse group of birds (*Seibold & Helbig, 1995*; *Wink, 1995*; *Johnson et al., 2016*). Old World vultures are descendants of the Accipitridae family, which also comprise eagles, hawks, kites and buzzards (*Wink, 1995*; *Lerner & Mindell, 2005*; *Clements et al., 2019*). Old and New World vultures belong to two different families, Accipitridae and Cathartidae, respectively (*Wink, 1995*; *Clements et al., 2019*). According to Clements' taxonomic classification, Accipitridae, which belongs to the order Accipitriformes, is one of the largest non-passerine families comprising 252 species (227 are monophyletic while 25 are polyphyletic) (*Clements et al., 2019*). The order Accipitriformes also includes the families Sagittariidae, Pandionidae and Cathartidae, although the latter is now classified under the order Cathartiformes (*Wink, 1995*; *Clements et al., 2019*).

Raptors are primarily predator birds that hunt and kill their prey but also include carrion feeders. These include members of the Accipitridae, Falconidae, Cathartidae, Strigidae and Tytonidae families (*Clements et al., 2019*). Raptors are valuable indicators of habitat quality based on their ecological sensitivity as predators and scavengers (*Lerner & Mindell, 2005*). However, vultures belonging to the Accipitriformes order are currently facing devastating drops in their population numbers from an array of problems ranging from loss of their natural habitat, collision with high-tension electric cables and wind turbines, intentional poisoning of animal carcasses by poachers of endangered wildlife species and accidental ingestion of carcasses contaminated with pharmaceuticals (*Ogada et al., 2016*; *Naidoo et al., 2018*; *Adawaren et al., 2018*). One notable incident was the near complete extinction of three *Gyps* vulture species the white-rumped vulture (*Gyps bengalenesis*), the Indian vulture (*G. indicus*) and the slender-billed vulture (*G. tenuirostris*) in India, Nepal and Pakistan from the consumption of carcasses of animals dosed with diclofenac (*Oaks et al., 2004*; *Swan et al., 2006b*; *Naidoo et al., 2009*; *Adawaren et al., 2018*). At present, it is estimated that the drug caused the deaths of over 10 million vultures in the region (*Naidoo et al., 2018*). Furthermore, diclofenac has been implicated as the cause of death of a steppe eagle (*Aquila nipalensis*), a member of the Accipitridae family with classical sign of toxicity seen in vultures (*Sharma et al., 2014*). This incidence raises concern on the vulnerability of the Accipitriformes and other raptors to diclofenac toxicity. Complications due to the toxic consequences of diclofenac have also been reported in the Cape vulture (*Swan et al., 2006b*; *Naidoo et al., 2009*; *Naidoo et al., 2010*; *Naidoo et al., 2018*; *Adawaren et al., 2018*).

Diclofenac, a non-steroidal anti-inflammatory drug (NSAID) mostly understood for its analgesic, anti-inflammatory and an antipyretic characteristic, is used for the treatment of inflammatory disorders in humans and animals. The drug was, however, sufficient to cause death within 48 h of exposure in vultures that had fed on the carcasses in which the drug was present, with signs of renal failure associated with visceral and articular gout being evident on necropsy (*Oaks et al., 2004*; *Swan et al., 2006b*; *Naidoo & Swan, 2009*). While the general mechanism of action of the NSAIDs and their toxicity in mammals is well characterized, the same cannot be said for vultures. Despite the exact cellular mechanism underlying toxicity remaining unknown, the clinical progression of toxicity is well established (*Swan et al., 2006b*; *Naidoo et al., 2009*). Vultures generally show signs of depression as early as 24 h after exposure (depression characterized by head drooping, reluctance to move,

open wings, decreased appetite, loss of aggression, etc.). At approximately 48 h post exposure, the affected animals usually died, with post-mortem showing characteristic gross lesions of visceral and articular gout and histological lesions of renal tubular necrosis, especially the proximal convoluted tubules (PCT) of the kidney and hepatocytes of the liver (*Swan et al., 2006b*; *Naidoo et al., 2009*). At the physiological level, the drug is associated with massive elevation in plasma uric acid amount, acidaemia, hyperkalaemia and increases in plasma liver enzyme activity. In terms of the temporal relationship, the first signs of depression correspond to the first elevation in uric acid amount indicative of early kidney damage, followed by increase in plasma liver enzyme activities indicative of hepatic necrosis, and, lastly, death associated with hyperkalaemia and acidosis. While speculative, the increase in potassium is purported to induce cardiac failure and death (*Naidoo et al., 2007*).

Following the discovery of diclofenac as the cause for these vulture deaths, research has shown that these Old World bird species are also vulnerable to other NSAIDs toxicity, with the notable exception being meloxicam thus far (*Swan et al., 2006a*; *Naidoo et al., 2010*; *Adawaren et al., 2018*). The scenario is, however, different with the New World vultures. In a study in which the Turkey vulture (*Cathartes aura*) was exposed to diclofenac 25 mg/kg, a concentration that was 100 times above the median lethal dose (0.1–0.2 mg/kg) in Old World vultures, no overt toxicity was evident. Furthermore, the diclofenac could hardly be detected in the tissue after necropsy, with the concurrent pharmacokinetics study demonstrating a low plasma half-life of elimination of 6 h, in comparison to 12–16 h observed in Cape vultures (*Rattner et al., 2008*; *Naidoo et al., 2009*). The high sensitivity of the Old World vultures also contrasts with other bird species whereby high doses in the region of 10 mg/kg was needed to induce toxicity in chickens (*Gallus gallus*), with a corresponding plasma half-life elimination predicted within the range of 14 h in domestic chicken. The Pied crow (*Corvus albus*) was less sensitive with no signs of toxicity at 10 mg/kg and a plasma half-life of 2.5 h (*Naidoo et al., 2007*; *Naidoo et al., 2010*).

Due to inter-species sensitivity and the apparent relationship between the plasma half-life of elimination, it was suggested that the lethal effect of the NSAIDs in avian species is associated with their ability to metabolise the drug in species-specific manner. From work in other species, this limitations tends to be an evolutionary link in the Cytochrome P450 enzyme network, which is responsible for xenobiotic metabolism. *Naidoo et al. (2010)* postulated that toxicity in vultures was due to zero-order metabolism related to a possible evolutionary pharmacogenetic defect in the CYP2C family resulting in non-expression of the enzyme system, based on similar effects in human with metabolic defects in the same enzyme family (*Naidoo et al., 2010*). The CYP enzyme of the vulture has yet to be identified. As CYP enzymes share an evolutionary link, we speculate that species susceptible to toxicity are closely related, which might be visualized using a phylogeny derived from complete mitogenomes (*Bort et al., 1999*; *Goodman and Gilman, 2011*; *Watanabe et al., 2013*; *Yang, Ye & Huang, 2016*).

## MATERIALS AND METHODS

### Materials

Sodium pentobarbital (Euthapent®), ZR Genomic DNA Isolation kit (Zymo Research), BigDye Terminator sequencer Cycle Sequencing Kit, oligonucleotide primers (Integrated DNA Technologies), liquid nitrogen (Afrox), 2 ml cyrotubes (Greiner Bio-One, Frickenhausen) were used in the study. The equipment used for the study were −80 °C refrigerator, NanoDrop spectrophotometer (Thermo Fisher Scientific), microcentrifuge (Eppendorf), ION Torrent S5 (Thermo Fisher Scientific) Next Generation Sequencer, 540 ION chip (Thermo Fisher Scientific), ABI 3130 Genetic Analyser (Applied Biosystems) and SimpliAmp Thermal Cycler (Thermo Fisher Scientific).

### Methods

#### Publicly available sequence information

The complete mitochondrial genomes (*mtDNA*) of the bird species used for these studies were obtained from GenBank. The species belonged to six families of raptors, namely, Accipitridae, Pandionidae, Sagittariidae, Cathartidae, Falconidae and Strigidae respectively (Table 1).

#### Collection of skin samples and genomic DNA extraction

This study was authorized by the Animal Ethics Committee of the University of Pretoria, South Africa, with project numbers V093-15 and V097-17 in 2015 to 2017. Samples were opportunistically collected immediately after the euthanasia of an individual Cape vulture (*Gyps coprotheres*) with intravenous injection of pentobarbitone for a non-treatable physical injury. Skin samples were stored in sterile cryotubes and placed immediately into liquid nitrogen (−196 °C) for 10 min to snap-freeze the samples, which were then transferred into the −80 °C refrigerator until genomic DNA extraction. The frozen Cape vulture skin samples were allowed to melt at ambient temperature and approximately 25 mg of the thawed tissue was excised for DNA extraction using the ZR Genomic DNA Isolation kit (Zymo Research) according to manual instructions. The quality of the extracted sample was examined using a NanoDrop reader.

#### Genome sequencing

Genome sequencing was performed at the Uppsala Genome Centre, Uppsala University, Uppsala, Sweden, on the ION S5 XL platform (Thermo Fisher. 2015). The genome sequencing was conducted according to manual instruction. The run was performed on 200 bp read length chemistry on an ION-540 chip

#### Mitogenome assembly and annotation

The NGS sequence quality was evaluated using the FastQC software (*Bioinformatics, 2011*). Based on the quality assessment, the data was trimmed using the Trimmomatic program (*Bolger, Lohse & Usadel, 2014*) and the total dataset was down sampled to 10 million reads. Thereafter, reads with lengths exceeding 100 bp were selected for subsequent assembly. The de-novo assembly was conducted using the CLC Genomic Workbench version 6.0 software on the default settings. The subsequent assemblies were used to create a database against
**Table 1** **Mitogenome accession number and Diclofenac Toxicity Status of Bird Species investigated in this study as classified by** *Clements et al. (2019)*. (A) Bird species names, (B) Family name, (C) Accession number, (D) Genera Diclofenac toxicity status, (E) References.

| Species | Family | Accession number |
| --- | --- | --- |
| *Accipiter gentilis* (Northern Goshawk) | Accipitridae | NC_011818 |
| *Accipiter nisus* (Eurasian Sparrowhawk) | Accipitridae | NC_025580 |
| *Accipiter soloensis* (Chinese Sparrowhawk) | Accipitridae | KJ680303 |
| *Accipiter virgatus* (Besra) | Accipitridae | NC_026082 |
| *Aegypius monachus* (Cinereous Vulture) | Accipitridae | KF682364.1 |
| *Aquila chrysaetos* (Golden Eagle) | Accipitridae | NC_024087 |
| *Aquila fasciata* (Bonelli's Eagle) | Accipitridae | KP329567 |
| *Aquila heliacal* (Eastern imperial eagle) | Accipitridae | NC_035806 |
| *Buteo buteo* (Common Buzzard) | Accipitridae | NC_003128 |
| *Buteo buteo burmanicus* (Himalayan) | Accipitridae | KM364882 |
| *Buteo fasciatus* (Bonelli's eagle) | Accipitridae | NC_029188 |
| *Buteo hemilasius* (Upland Buzzard) | Accipitridae | NC_029377.1 |
| *Buteo lagopus* (Rough-legged Hawk) | Accipitridae | KP337337 |
| *Butastur indicus* (Grey-faced Buzzard) | Accipitridae | NC_032362 |
| *Butastur liventer* (Rufous-wing Buzzard) | Accipitridae | AB830617 |
| *Cathartes aura* (Turkey Vulture) | Cathartidae | NC_007628 |
| *circus cyaneus* (Northern Harrier) | Accipitridae | KX925606 |
| *circus melanoleucos* (Pied Harrier) | Accipitridae | NC_035801 |
| *Gyps coprotheres* (Cape vulture) | Accipitridae | MF683387 |
| *Gyps fulvus* (Griffon vulture) | Accipitridae | NC_036050 |
| *Gyps himalayensis* (Himalayan vulture) | Accipitridae | KY594709.1 |
| *Nisaetus alboniger* (Blyth's Hawk-Eagle) | Accipitridae | NC_007599 |
| *Nisaetus nipalensis* (Mountain Hawk-Eagle) | Accipitridae | NC 007598.1 |
| *Spilornis cheela* (Crested Serpent-Eagle) | Accipitridae | NC_015887 |
| *Sagittarius serpentarius* (Secretary-bird) | Sagittariidae | NC_023788 |
| *Pandion haliaetus* (Osprey) | Pandionidae | NC_008550 |
| *Strix leptogrammica* (Brown wood owl) | Strigidae | KC953095.1 |
| *Falco columbarius* (Merlin) | Falconidae | KM264304.1 |
| *Falco tinnunculus* (Common Kestrel) | Falconidae | EU196361.1 |
| *Falco sparverius* (American kestrel) | Falconidae | DQ780880.1 |
| *Falco naumanni* (Lesser Kestrel) | Falconidae | KM251414.1 |
| *Falco peregrinus* (Barbary falcon) | Falconidae | JQ282801.1 |
| *Falco cherry* (Cherry Falcon) | Falconidae | KP337902.1 |
| *Falco rusticolus* (Gyrfalcon) | Falconidae | KT989235.1 |

which a representative mitochondrial genome was queried. The contig with significant similarity to the query across its entire length was then submitted to the MITOS server (*Bernt et al., 2013*) in order to perform annotation of the mitochondrial genome.

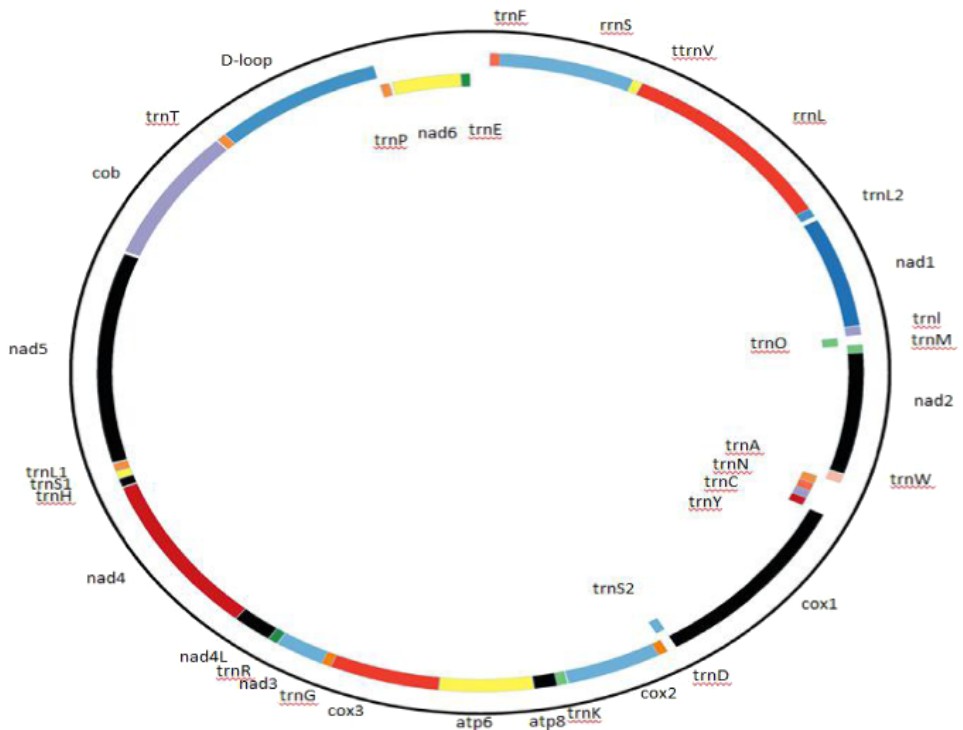

**Figure 1 Complete mitochondrial genome organization and mitogene arrangement of *Gyps coprotheres*.** Genes found on the coding strand are indicated outside the mitochondrial genome map, while the mitogenes coded on the complementary strand are indicated inside the map.

## Mitogenome structure, organization and characterization of *Gyps coprotheres*

The Cape vulture mitogenome order, organization and characterization were described as presented in Fig. 1 and Table 2. Gene overlap and intergenic-space sequences were determined manually. The putative origin of light-strand replication ($O_L$) and control region were identified by comparison with the homologous sequences of other bird species from the Accipitriformes order.

## Phylogenetics

The mitogenome phylogeny was inferred using maximum likelihood analysis model in MEGA X between bird species included in this study (*Hall, 2013*). The raptor species included in the phylogenetic analysis belong to the Accipitridae, Falconidae, Strigidae and Cathartidae families while *Strix leptogrammica* was used as an outgroup (Fig. 2). To determine the evolutionary relationship among the raptor bird species, each bird species complete mitogenome was analysed using the maximum likelihood method in MEGA X. The model test feature in MEGAX was used to evaluate the best-fit model of evolution, of which GTR + G + I was determined to be the best for constructing the phylogenetic tree. The preference model GTR + G + I was derived using the Molecular Evolutionary Genetic Analysis (MEGA) software by conducting DNA/Protein models analysis to determine the Disparity Index Test Pattern of heterogeneity of the aligned nucleotide sequences used for

**Table 2 Characteristics of the mitochondrial genome of *Gyps coprotheres*.** (A) Mitogene names, (B) mitogene position, (C) Mitogene nucleotide size, (D) Mitogene start and stop codon, (E) Mitogene intergenic overlap, (F) Mitogene strand, (G) Mitogene A + T% nucleotide composition.

| Gene | Position | | Size | Condon | | Intergenic overlap | Strand | Nucleotide composition |
|------|------|----|------|-------|------|---------|--------|-----------|
| | From | To | Nucleotide | Start | Stop | | | A + T% |
| tRNA-Phe | 1 | 70 | 70 | | | 0 | H | 47.2 |
| 12S rRNA | 70 | 1037 | 968 | | | 0 | H | 51.2 |
| tRNA-Val | 1037 | 1108 | 72 | | | 0 | H | 55.6 |
| 16S rRNA | 1109 | 2709 | 1,601 | | | 1 | H | 54 |
| tRNA-Leu | 2710 | 2783 | 74 | | | 1 | H | 47.3 |
| ND1 | 2822 | 3800 | 978 | ATG | AGG | 39 | H | 53.9 |
| tRNA-Ile | 3769 | 3840 | 72 | | | −31 | H | 55.5 |
| tRNA-Gln | 3854 | 3924 | 71 | | | 14 | L | 67.6 |
| tRNA-Met | 3924 | 3992 | 69 | | | 0 | H | 49.3 |
| ND2 | 3993 | 5039 | 1,047 | ATG | TAG | 1 | H | 52.6 |
| tRNA-Trp | 5038 | 5109 | 72 | | | −1 | H | 62.5 |
| tRNA-Ala | 5111 | 5179 | 69 | | | 2 | L | 56.5 |
| tRNA-Asn | 5182 | 5254 | 73 | | | 3 | L | 50.7 |
| tRNA-Cys | 5257 | 5323 | 67 | | | 3 | L | 49.3 |
| tRNA-Tyr | 5324 | 5393 | 70 | | | 1 | L | 55.7 |
| COX1 | 5395 | 6945 | 1,551 | GTG | AGG | 2 | H | 52.8 |
| tRNA-Ser | 6937 | 7010 | 74 | | | −8 | L | 52.7 |
| tRNA-Asp | 7015 | 7083 | 69 | | | 5 | H | 59.4 |
| COXII | 7086 | 7769 | 684 | ATG | TAA | 3 | H | 52.8 |
| tRNA-Lys | 7771 | 7841 | 71 | | | 2 | H | 59.1 |
| ATP8 | 7843 | 8010 | 168 | ATG | TAA | 2 | H | 55.6 |
| ATP6 | 8001 | 8684 | 684 | ATG | TAA | −9 | H | 54.8 |
| COXIII | 8684 | 9467 | 784 | ATG | T | 0 | H | 52.9 |
| tRNA-Gly | 9468 | 9536 | 69 | | | 1 | H | 66.6 |
| ND3 | 9537 | 9710 | 351 | ATT | TAA | 1 | H | 55.1 |
| tRNA-Arg | 9893 | 9961 | 69 | | | 183 | H | 59.4 |
| ND4L | 9963 | 10259 | 297 | ATG | TAA | 2 | H | 53.9 |
| ND4 | 10253 | 11630 | 1,378 | ATG | T | −6 | H | 51.3 |
| tRNA-His | 11631 | 11700 | 70 | | | 1 | H | 65.7 |
| tRNA-Ser | 11702 | 11766 | 65 | | | 2 | H | 55.4 |
| tRNA-Leu | 11767 | 11837 | 71 | | | 1 | H | 62.0 |
| ND5 | 11847 | 13652 | 1,806 | ATA | TAA | 10 | H | 55.2 |
| Cytb | 13665 | 14807 | 1,143 | ATG | TAA | 13 | H | 52.3 |
| tRNA-Thr | 14810 | 14877 | 68 | | | 3 | H | 64.8 |
| tRNA-Pro | 16083 | 16152 | 70 | | | 1206 | L | 61.4 |
| ND6 | 16174 | 16692 | 519 | ATG | TAG | 22 | H | 50.3 |
| tRNA-Glu | 16693 | 16763 | 71 | | | 1 | L | 62.0 |
| D-loop | 14878 | 16082 | 4 | | | −1885 | H | 58.6 |
| Unknown Region | 16764 | 16908 | 145 | | | 682 | H | 62.7 |

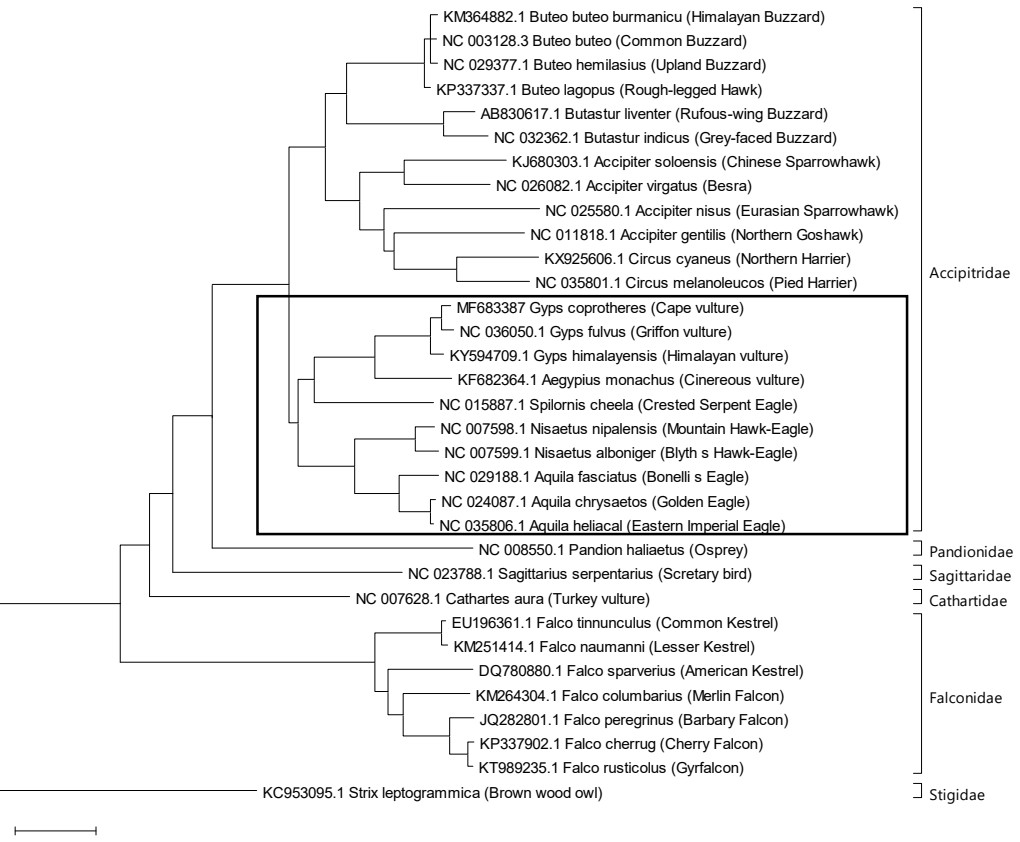

**Figure 2** **Phylogenetic Analysis Result using complete mitogenome.** Results of Phylogenetic analyses using maximum likelihood (ML) analysis indicated evolutionary relationships among 33 raptor species based on complete mitogenome sequences. *Strix leptogrammica* was used as outgroup. Bootstrap support values for ML analyses are indicated on the nodes. The solid border rectangle indicates a close phylogenetic relationship between Old World vultures, Hawk and Eagles confirmed with high bootstrap values with greater chances of shared susceptibility based on their close evolutionary relationship. While distantly related raptor bird species are less likely to succumb to diclofenac toxicity as reported for Turkey vulture (*Cathartes aura*) (*Rattner et al., 2008*).

the construction of a maximum likelihood phylogenetic tree. This model was implemented in MEGAX to construct phylogenetic trees using the maximum likelihood algorithm by performing 1000 bootstrap replicates according to the protocol described by *Hall (2013)*.

# RESULTS

## Mitogenome structure, organization and characterization of *Gyps coprotheres*

The Cape vulture mitogenome is a 16,908 bp circular DNA molecule with 13 protein coding genes (PCGs), 22 transfer RNA (tRNA), 2 ribosomal RNA (rRNA) and a non-coding region known as the *D-loop* (GenBank accession no. MF683387; Fig. 1, Table 2). The most used start codon is ATG with 76.92% frequency while ATA, GTG and ATT were alternate initiation codons. On the other hand, TAA was the most used termination code with

53.85% frequency, while AGG and TAG served as alternate stop codons. Furthermore, NAD4 and COX3 do not have stop codons, but had T as their last nucleotide (Table 2). The architecture of the Cape vulture *mtDNA* was similar to those of the raptor bird species included in this study (*Jiang et al., 2019*).

### Phylogenetic analyses

For this study, the evolutionary relationships among the bird species were investigated using phylogenetic analysis of the complete mitogenome among all the raptor species whose complete mitochondrial genome are available in GenBank. The choice of the mitogenomes as surrogates to investigate evolutionary relationships was because the mitogenome is composed of unique DNA sequences with evolutionary characteristic among animal species (*Jiang et al., 2015*; *Jiang et al., 2019*).

## DISCUSSION

The phylogenetic trees inferred from the complete mitogenome sequences among the raptor bird species included in this study indicated a monophyletic relationship among the Accipitridae. These monophyletic relationships were presented as clusters of Buteo + Butastur, Accipiter + Circus Gyps + Aegypius + Spilornis and Nisaetus + Aquila clades, respectively, with 100% bootstrap values (Fig. 2). Also the tree indicates that the *Gyps* species are closely related to *Aegypius* followed by *Spilornis*, *Nisaetus*, and *Aquila* genera respectively. Furthermore, the *Pandion* genus had close relationship to the Accipitridae followed by *Sagittarius, Cathartes* and the *Falco* genera, respectively (*Johnson et al., 2016*). In addition, the result showed a monophyletic relationship among falcons, which are closely related to Accipitridae compared to owls. The hierofalcons (*Falco cherrug* and *Falco rusticolus*) and *Falco peregrinus* cluster together into one clade with 100% bootstrap support values (Fig. 2). Furthermore, *Falco tinnunculus* and *Falco naumanni* had a monophyletic relationship with 100% bootstrap value, while *Falco columbarius* and *Falco sparverius* were outgroups. This study further confirmed the monophyletic relationship existing among falcons (*Helbig et al., 1994*; *Wink et al., 2004*; *Nittinger et al., 2005*; *Doyle et al., 2018*).

Earlier pharmacokinetics studies of NSAIDs in Old World vultures and other bird species indicated that there is a species-specific relationship associated to NSAIDs toxicity among bird species (*Swan et al., 2006b*; *Rattner et al., 2008*; *Naidoo et al., 2009*; *Naidoo et al., 2010*; *Naidoo et al., 2011*; *Naidoo et al., 2018*). In addition, the detrimental consequences of diclofenac have also been reported in the steppe eagle, a member of the *Aquila* genus (*Sharma et al., 2014*).

The phylogenetic analysis results (Fig. 2) clustered the Old World vultures (*Gyps* and *Aegypius* genera), hawks (*Nisaetus* genus) and eagles (*Spilornis* and *Aquila* genera) into the same clade. Considering the susceptibility of Old World vulture and eagle (*Aquila nipalensis*) species to diclofenac toxicity, it is possible that these closely-related bird species could also be sensitive to diclofenac toxicity (Fig. 2). The hawk and eagle families are of more particular concern than the other members of Accipitridae. However hawks and eagles may also be somewhat more protected as a result of their feeding habits which solely rely on hunting and less commonly on carrion feeding as opposed to the Old World

vultures which are purely carrion feeders i.e., the opportunity to be exposed to medicated carrions is higher for the vultures. Thus, it would be important to ascertain these species sensitivity as already undertaken in the pied crow and Turkey vultures. Based on this premise, the results also tend to suggest that falcons, owls, harries, bustards, buzzards, kestrels and hawks would be less likely susceptible to diclofenac, since they are an out grouping as seen with the Turkey vulture which is resistant to diclofenac at a dose that is 100 times the lethal dose seen in Old World vultures.

The possible reason for a shared susceptibility among Accipitridae may be explained by evolutionary changes in the cytochrome P450 (CYP) group of enzymes. These enzymes are important in the detoxification of environmental pollutants and xenobiotics (*Bort et al., 1999*; *Goodman and Gilman, 2011*; *Watanabe et al., 2013*). It thus stands to reason that, with the mitogenome indicating species similarity, these species would evolve under similar environmental conditions and thus develop similar CYP enzyme capacity. This was demonstrated with the cholinesterase enzyme system in which the concentrations in herbivores is naturally higher than carnivores, due to plants having higher concentrations of natural acetyl choline like substances in comparison to animals (*Ruiz-Garcia et al., 2008*). As a result, the evolutionary adaption of higher enzyme concentration of these enzyme results in herbivores being less susceptible to organophosphorus toxicity (*Ruiz-Garcia et al., 2008*).

Evolutionary variations in the CYP450 enzymes between diclofenac resistant bird species (Turkey vulture, pied crow, chicken) and susceptible Old World vultures can better explain species-specific toxicity observed among avian. It is therefore imperative to identify members of the CYP450 group of genes in birds to fully elucidate the reason behind resistance and susceptibility to NSAIDs.

## CONCLUSION

The architecture of the Cape vulture mitogenome was similar to the raptor bird species included in this study. The mitogenome phylogenetic analyses suggest the possibility of sensitivity to diclofenac toxicity among hawks and eagles.

## ACKNOWLEDGEMENTS

Kerri Wolter of VulPro is thanked for making the vulture tissue available.

### Funding

Sequencing was sponsored by Thermo Fisher Scientific (Mr Wayne Barnes). Emmanuel Adawaren was on a bursary sponsored by the National Research Foundation (NRF) of South Africa (Grant no. 87772). The funders had no role in study design, data collection and analysis, decision to publish, or preparation of the manuscript.

## Grant Disclosures

The following grant information was disclosed by the authors:

Thermo Fisher Scientific (Mr Wayne Barnes).

National Research Foundation (NRF) of South Africa: 87772.

## Competing Interests

The authors declare there are no competing interests.

## Author Contributions

- Emmanuel Oluwasegun Adawaren conceived and designed the experiments, performed the experiments, analyzed the data, prepared figures and/or tables, authored or reviewed drafts of the paper, and approved the final draft.
- Morne Du Plessis and Lillian Mukandiwa analyzed the data, authored or reviewed drafts of the paper, and approved the final draft.
- Essa Suleman and Vinny Naidoo analyzed the data, prepared figures and/or tables, authored or reviewed drafts of the paper, and approved the final draft.
- Duodane Kindler performed the experiments, prepared figures and/or tables, and approved the final draft.
- Almero O. Oosthuizen analyzed the data, prepared figures and/or tables, and approved the final draft.

## Animal Ethics

The following information was supplied relating to ethical approvals (i.e., approving body and any reference numbers):

The bird species used during this study was authorised by the Animal Ethics Committee of the University of Pretoria, South Africa (#V093-15 and #V097-17).

## DNA Deposition

The following information was supplied regarding the deposition of DNA sequences:

The data is available at GenBank: MF683387.

## Data Availability

The data is available at GenBank: PRJNA389655.

## Supplemental Information

Supplemental information for this article can be found online at http://dx.doi.org/10.7717/peerj.10034#supplemental-information.

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
