# Peer review of "The complete mitochondrial genome of Gyps coprotheres (Aves, Accipitridae, Accipitriformes): phylogenetic analysis of mitogenome among raptors"

_PeerJ, doi:10.7717/peerj.10034_

## Round 0.1 · original submission · Major Revisions

Dear Dr. Adawaren and colleagues:

Thanks for submitting your manuscript to PeerJ. I have now received two independent reviews of your work, and as you will see, the reviewers raised some substantial concerns about the research. Despite this, these reviewers are optimistic about your work and the potential impact it will have on research studying old world and molecular correlates with lethality induced by diclofenac. Thus, I encourage you to revise your manuscript, accordingly, taking into account all of the concerns raised by both reviewers.

Please also not that reviewer 1 has included a marked-up version of your manuscript.

In your revision, please provide the missing materials noted by the reviewers, and also consider the missing references that have been noted. Please address the grammatical problems. Also, please address the concern for character mapping, and why some of the other species are not discussed in your work.

I look forward to seeing your revision, and thanks again for submitting your work to PeerJ.

Good luck with your revision,

-joe

·

Basic reporting

Emmanuel et al. explored the complete mitochondrial genome of Gyps coprotheres and investigated the correlation between phylogeny and susceptibility to diclofenac toxicity among raptors. The manuscript is interesting and fitted for Peer J. however there is some problems for example we did not found the table and figure Legends, and some new references did not cite. In addition, I suggested the authors may improve in introduction and discussion part to make more concise. Some minor and specific suggestion is attached in the manuscript.

Experimental design

suggested to add new GenBank accession numbers:KY594709 into your analysis

Validity of the findings

meet the standards

Additional comments

Emmanuel et al. explored the complete mitochondrial genome of Gyps coprotheres and investigated the correlation between phylogeny and susceptibility to diclofenac toxicity among raptors. The manuscript is interesting and fitted for Peer J. however there is some problems for example we did not found the table and figure Legends, and some new references did not cite. In addition, I suggested the authors may improve in introduction and discussion part to make more concise. Some minor and specific suggestion is attached in the manuscript.

Reviewer 2 ·

Basic reporting

The language is professional (with a few exceptions, see below), as is the article structure.

However, the results presented are not suitable for testing the hypothesis. The authors’ aim is to test whether diclofenac toxicity is correlated with relatedness amongst raptor species (with particular emphasis on new and old world vultures). However, the authors do not provide information on diclofenac susceptibility in all 32 species included in their phylogeny. Instead, they review diclofenac toxicity in eight species (by my count). It is unclear whether this information was just omitted from the manuscript (in which case it should be included in a table in the next draft) or does not exist. If diclofenac susceptibility is only well understood for eight species, I do not believe that any meaningful conclusions can be reached.

Minor comments:

Line 37: “Rejuvenating” is not an appropriate word here.

Line 87: Which three species are you referring to, explicitly?

Lines 287-288: The authors argue that “all old world vultures are vulnerable to the lethal consequences of diclofenac”. If this is true, I would provide a table with each of the 32 species in the phylogeny and a citation describing diclofenac toxicity.

Lines 261-283: The authors should review their results in the context of what is already known regarding raptor phylogenetics. For example, you might refer to Nittinger et al. 2005, Helbig et al. 1994, Wink et al. 2004 and Doyle et al. 2018 when discussing relatedness amongst falcon species.

References:

Nittinger F, Haring E, Pinsker W, Wink M, Gamauf A. Out of Africa? Phylogenetic relationships between Falco biarmkus and the other Hierofalcons (Aves: Falconidae). J Zool Syst Evol Res. 2005;43:321–31.

Helbig AJ, Seibold I, Bednarek W, Gaucher P, Ristow D, Scharlau W, Schmidl D, Wink M. 1994. Phylogenetic relationships among Falcon species (genus Falco) according to DNA sequence variation of the cytochrome b gene. In: Meyburg B-U, Chancellor RC, editors. Raptor Conservation Today. Berlin: World Working Group Birds of Prey and Pica Press; 1994. p. 593-599.

Wink M, Sauer-Gurth H, Ellis D, Kenward R. Phylogenetic relationships in the Hierofalco complex (Saker-, Gyr-, Lanner-, Laggar Falcon). In: Raptors Worldwide; 2004. p. 499–504.

Doyle JM, Bell DA, Bloom PH, Emmons G, Fesnock A, Katzner TE, LaPre L, Leonard K, SanMiguel P, Westerman R, DeWoody JA. 2018. New insights into the phylogenetics and population structure of the prairie falcon (Falco mexicanus). BMC Genomics 19:233.

Experimental design

The research question is relevant and meaningful. However, the investigation is not rigorous. As noted above, the authors would need data on diclofenac toxicity in a much larger number of species. Although it is not my area of expertise, it also seems appropriate to use some form of phylogenetic/character mapping.

Minor comments:

Lines 177-180: Can you describe the library preparation?

Lines 187-190: I’m not following you here. You generated multiple assemblies of the mitochondrial genome? What is the “representative mitochondrial genome queried”?

Line 202 Italicize Gyps coprotheres?

Line 214: Why generate three different phylogenetic trees rather than a single tree using the complete mitogenomes?

Validity of the findings

I do not believe the analysis to be robust (see above). If the comments above cannot be addressed, I believe it might be best to revise the manuscript as a note for Mitochondrial DNA Part B, without the emphasis on diclofenac toxicity.

Minor comments:

Figure 1 appears distorted?

---

## Round 0.2 · Minor Revisions

Dear Dr. Adawaren and colleagues:

Thanks for resubmitting your revision and addressing the concerns of the reviewers. Reviewer 1 has a minor concern; however, Reviewer 2 has provided a second review of your work and new concerns have arisen. Please address these as soon as possible and resubmit your work. I am sure your manuscript will be much closer to acceptance once these new concerns are addressed.

Please consider dropping the emphasis on diclofenac toxicity, or at least minimizing the correlative nature with respect to phylogeny. Your study is not very robust to support this claim.

I look forward to seeing your revision, and thanks again for submitting your work to PeerJ.

Good luck with your revision,

-joe

·

Basic reporting

no comment

Experimental design

no comment

Validity of the findings

no comment

Additional comments

i only have minor suggestion see the attach file

Reviewer 2 ·

Basic reporting

Lines 132-135: Might revise to: “As CYP enzymes share an evolutionary link, we speculate that species susceptible to toxicity are closely related. This might be visualized using a phylogeny derived from complete mitogenomes.” The line “For this study the complete mitogenomes were used as surrogates for these comparisons” (lines 135-136) could then be removed.

The authors noted in their rebuttal letter that they revised the phylogenetic tree to make use only of complete mitogenomes, but there are a number of references to the trees built with mitochondrial genes built into the manuscript (e.g., Lines 163-165: “In addition, blood samples were collected from six unrelated Cape vulture in polycarbonated heparinized tubes for PCR amplification of COX1, COX3 and NAD3 genes.” The COX1 phylogeny, etc. are also still included in the supplemental files and Table 2 seems unnecessary).

Line 250: Please revise to correct grammar: “While the Gyps+Aegypius+Spilornis and Nisaetus+Aquila genera forming a clade."

Lines 271-283: There are a number of grammatical errors throughout this paragraph (e.g., double period on line 273, missing parentheses in lines 282-283). I would review the entire paragraph and edit appropriately.

Figure 2: There appear to be several preliminary notes associated with Figure 2 that were left in my mistake (e.g., a note that the common name for Falco sparverius needs to be included in the figure).

Experimental design

The updated Table 1 indicates that the authors have susceptible/not susceptible statuses for 8 species (with citations). Only a single species is listed as being not susceptible. In a number of cases, the authors state that a species is “suspected” of being susceptible to diclofenac toxicity but there are no citations for this information included and I’m not sure where these arguments are coming from. If diclofenac susceptibility is only well understood for eight species, I do not believe that the authors can show a “correlation between phylogeny and susceptibility to diclofenac toxicity” (as argued in the rebuttal letter... and the title, which is concerning), particularly given that only a single species is described as being not susceptible! Personally, I do not believe this meets the journal’s requirement of “Rigorous investigation performed to a high technical & ethical standard”. That being said, the editor would know better than I the requirements of the journal, so I’ll leave it up to them.

Validity of the findings

I believe it might be best to revise the manuscript as a note for Mitochondrial DNA Part B, without the emphasis on diclofenac toxicity (see comments above).

---

## Round 0.3 · Minor Revisions

Dear Dr. Adawaren and colleagues:

Thanks for once again revising your manuscript and resubmitting your revision. I have received one review of your work, and as you will see, this reviewer still has some concerns. Please address these ASAP so we may move forward with accepting your work for publication.

Good luck with your revision,

-joe

Reviewer 2 ·

Basic reporting

The literature review and background/context seem sufficient and the article structure is professional. The mitogenome has been archived with GenBank, although it is not clear to me if the raw sequencing reads have been archived and/or if that is required by the journal.

There are some problems with punctuation/grammar throughout the manuscript, which seem to be associated with the most recent revisions. I am listing a few below but I think it would be useful if a PeerJ editor worked with the authors to polish the manuscript.

Lines 42-44: Revise to read: "Based on this, we speculate that other species could also be sensitive to the toxic effects of diclofenac, which warrants further investigation."

Lines 92-95: Please revise this line for clarity/grammar: "From work in other species, this limitations tends to be an evolutionary link in the Cytochrome P450 enzyme network, which is responsible for xenobiotic metabolism."

Lines 94-95: Revise to read: "As the CYP enzyme of the vulture has yet to be identified..."

Line 129: A period is needed at the end of the sentence.

Lines 170-171: Citation formatting for Jiang et al., 2015 and Jiang et al 2019 is not consistent, please revise to meet the specifications of the journal.

Line 177: Please revise to read: "Also the tree indicates...."

Lines 192: Please revise to read: "... it is possible that these closely-related bird species...."

Lines 205-206: Please revise to read: "These enzymes are important in the detoxification of environmental pollutants..."

Experimental design

The methods are described with sufficient detail, with the exception that I would include the following information included in the rebuttal letter but not in the manuscript (as far as I can tell): "The preference model GTR+G+I was derived using the Molecular Evolutionary Genetic Analysis (MEGA) software by conducting DNA/Protein models analysis to determine the Disparity Index Test Pattern of heterogeneity of the aligned nucleotide sequences used for the construction of a maximum likelihood phylogenetic tree."

Otherwise, I believe the experimental design meets the requirements of the journal.

Validity of the findings

I believe the findings/conclusions meet the requirements of the journal.

---

## Round 0.4 · accepted · Accept

Dear Dr. Adawaren and colleagues:

Thanks for revising your manuscript based on the concerns raised by the reviewers. I now believe that your manuscript is suitable for publication. Congratulations! I look forward to seeing this work in print, and I anticipate it being an important resource for groups studying old world vultures and molecular correlates with lethality induced by diclofenac. Thanks again for choosing PeerJ to publish such important work.

Best,

-joe